# A data-driven generative strategy to avoid reward hacking in multi-objective molecular design

Tatsuya Yoshizawa [1,2], Shoichi Ishida [1], Tomohiro Sato [2], Masateru Ohta[3], Teruki Honma[2] & Kei Terayama [1,4,5] ✉

Molecular design using data-driven generative models has emerged as a promising technology, impacting various fields such as drug discovery and the development of functional materials. However, this approach is often susceptible to optimization failure due to reward hacking, where prediction models fail to extrapolate, i.e., fail to accurately predict properties for designed molecules that considerably deviate from the training data. While methods for estimating prediction reliability, such as the applicability domain (AD), have been used for mitigating reward hacking, multi-objective optimization makes it challenging. The difficulty arises from the need to determine in advance whether the multiple ADs with some reliability levels overlap in chemical space, and to appropriately adjust the reliability levels for each property prediction. Herein, we propose a reliable design framework to perform multi-objective optimization using generative models while preventing reward hacking. To demonstrate the effectiveness of the proposed framework, we designed candidates for anticancer drugs as a typical example of multi-objective optimization. We successfully designed molecules with high predicted values and reliabilities, including an approved drug. In addition, the reliability levels can be automatically adjusted according to the property prioritization specified by the user without any detailed settings.

Molecular design using generative models has recently gained significant attention as a promising technology[1–3]. This approach has been applied to diverse fields, such as drug discovery[4–10] and the development of functional materials[11–19]. Such a design can be formulated as an inverse problem[1], i.e., designing molecules with desired properties. The key is to evaluate the designed molecules (calculate the reward function) using methods like molecular simulation or data-driven predictive modeling, to guide the design process. While simulation-based evaluation, grounded in physicochemical principles, is generally

accurate, it can be computationally expensive. Therefore, data-driven evaluation, based on supervised learning prediction models, is frequently used[2]. In addition, practical molecular design often requires the simultaneous optimization of multiple properties. This can be formulated as a multi-objective optimization problem[20], and molecular designs based on generative models and data-driven evaluations have been reported using this approach[4–7,9,10].

However, data-driven molecular design using prediction models faces the risk of reward hacking[21]. This phenomenon, well-known in

[1]Graduate School of Medical Life Science, Yokohama City University, 1-7-29, Suehiro-cho, Tsurumi-ku, Yokohama 230-0045 Kanagawa, Japan. [2]RIKEN Center for Biosystems Dynamics Research, 1-7-22, Suehiro-cho, Tsurumi-ku, Yokohama 230-0045 Kanagawa, Japan. [3]HPC- and AI-driven Drug Development Platform Division, RIKEN Center for Computational Science, 1-7-22, Suehiro-cho, Tsurumi-ku, Yokohama 230-0045 Kanagawa, Japan. [4]RIKEN Center for Advanced Intelligence Project, 1-4-1, Nihonbashi, Chuo-ku 103-0027 Tokyo, Japan. [5]MDX Research Center for Element Strategy, Institute of Science Tokyo, 4259, Nagatsuta-cho, Midori-ku, Yokohama 226-8501 Kanagawa, Japan. ✉e-mail: terayama@yokohama-cu.ac.jp

reinforcement learning and game AI, occurs when optimization deviates unexpectedly from the intended goals[22,23]. It arises when a reward function, used to guide optimization, produces unintended outputs due to inputs that deviate significantly from expected scenarios. A similar issue can arise in molecular design guided by a prediction model, leading to the design of molecules that diverge from the model's training data, resulting in reward hacking[21,24,25]. Consequently, the optimization process deviates from its intended direction. In such cases, the seemingly favorable predicted value of the designed molecule may be inaccurate. In fact, in drug design, there have been cases where unstable or complex molecules, distinct from existing drugs, have been designed due to reward hacking[24,26].

To address the issue of reward hacking, strategies have been developed to assess and improve the reliability of prediction models. The reliability of these models has long been formulated as applicability domains (ADs)[27–33] and has recently gained attention in the context of prediction uncertainty[34–40]. The AD is defined as "the response and chemical structure space in which the model makes predictions with a given reliability"[27], and predictions within the AD are expected to maintain a certain level of accuracy. The trade-off between

the size of an AD and the reliability of a prediction can be adjusted by changing the reliability level (Fig. 1a). A common approach to ensuring reliability is to guide the design of molecules within the AD of a single prediction model[41,42]. Additionally, a strategy for performing multi-objective optimization by considering prediction reliability has been reported, where multiple ADs are merged into a single AD (Fig. 1b), and molecular designs are then performed within this merged AD[9].

However, even when focusing on prediction reliability to mitigate reward hacking, maintaining reliability in multi-objective optimization is not straightforward. First, the strategy of merging multiple ADs is undesirable except in cases where multiple prediction models are trained on the same dataset. As prediction models have unique ADs depending on their training data, each AD should be considered separately when multiple prediction models are employed. In cases where multiple ADs are handled separately, as shown in the leftmost case of Fig. 1c, when ADs are defined for each prediction model at high reliability levels, they may not overlap if the training data for each model are distant in chemical space. In such situations, it is inherently difficult to generate molecules that fall within all ADs. As a compromise, it may be possible to overlap multiple ADs by lowering the

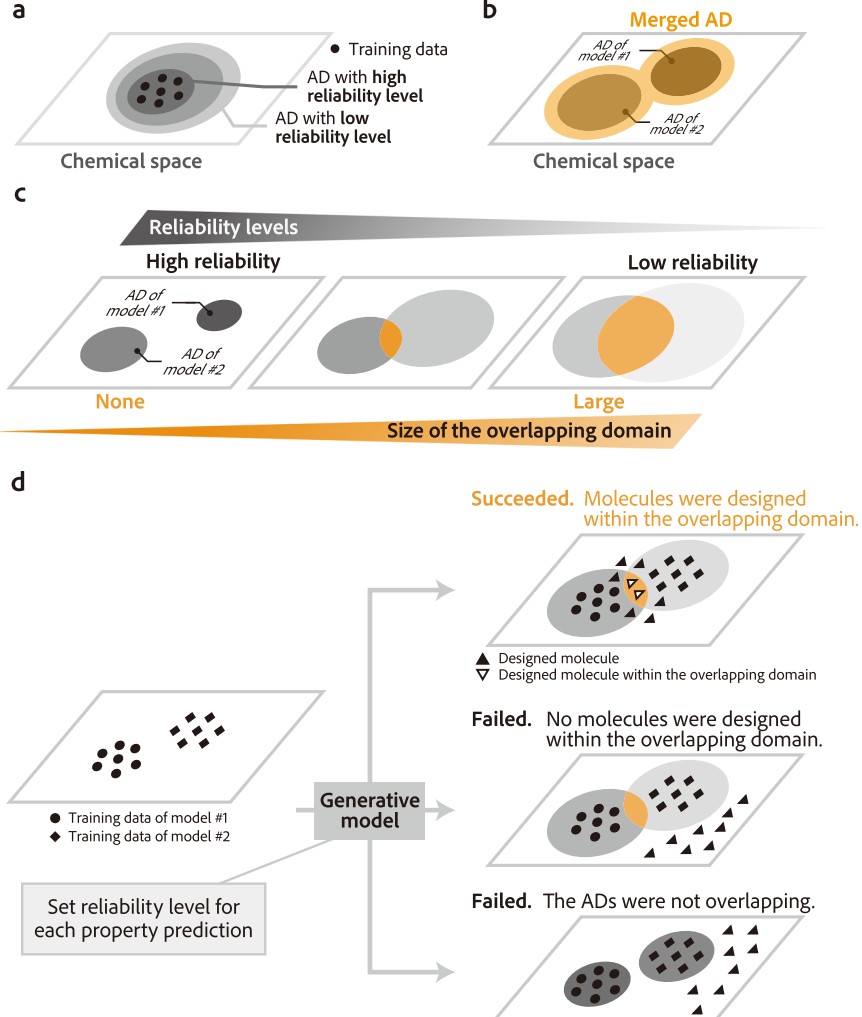

**Fig. 1 | The concept of applicability domain (AD) and difficulties of multi-objective optimization incorporating multiple ADs. a** The concept of AD. The size of an AD occupied in the chemical space can be varied by changing the reliability level. **b** One of the strategies for multi-objective optimization that incorporates ADs is to merge the ADs of multiple prediction models and regard them as a single AD. **c** The inherent trade-off between the reliability level and size of an overlapping domain (highlighted in orange) of multiple ADs in the context of multi-

objective optimization. **d** Examples of success and failure cases in molecular design incorporating ADs of multiple prediction models. Depending on the setting of the molecular design, multi-objective optimization may succeed in the overlapping region of multiple ADs (upper case). However, the molecular design may fall outside the overlapping region (middle case) and the ADs may not overlap each other depending on the reliability levels of the ADs (bottom case). Setting the appropriate reliability levels of the ADs before designing is difficult.

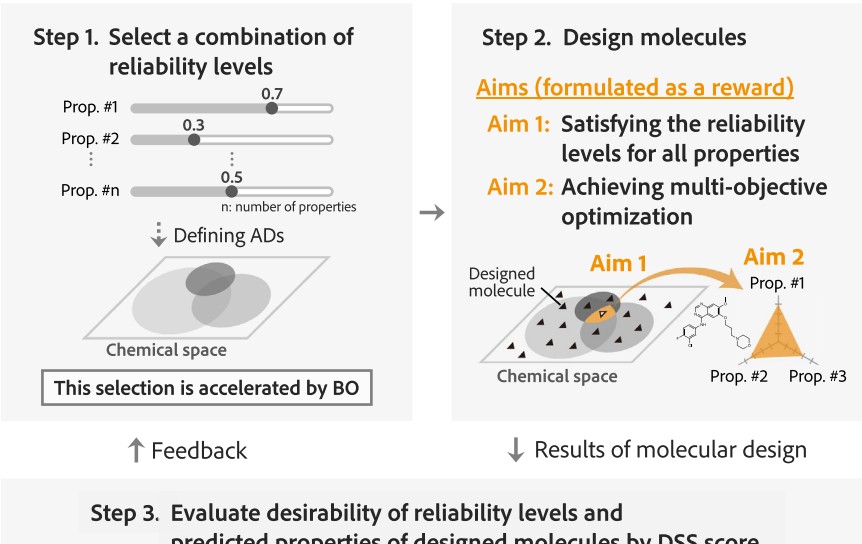

**Fig. 2 | Workflow of DyRAMO (Dynamic Reliability Adjustment for Multi-objective Optimization).** DyRAMO explores an appropriate combination of the reliability level for each target property prediction. The exploration is conducted by repeating the following three steps. In step 1, a reliability level is selected for each property prediction, and the applicability domains (ADs) of the prediction models for the properties are defined based on the selected reliability levels. In step 2, molecular design is performed with the aim of satisfying the reliability levels given in step 1 for all property predictions (Aim 1) and achieving multi-objective optimization (Aim 2). This design process is conducted to obtain molecules that fall within all defined ADs and have optimized target properties. In step 3, the evaluation of the molecular design is performed from two aspects: the desirability of reliability levels selected in step 1 and the degree of optimization of predicted properties of the molecules designed in step 2. The DSS (Degree of Simultaneous Satisfaction of prediction reliability and multiple property optimizations) score was introduced to make this assessment. By repeating this cycle, an appropriate combination of the reliability levels is searched for so as to improve the DSS score. In DyRAMO, Bayesian optimization (BO) is introduced to accelerate this exploration.

reliability levels (center and right cases in Fig. 1c). However, excessively low reliability levels may produce unreliable molecules, and thus, higher reliability levels are desirable. A further complication is the difficulty in determining the appropriate reliability level before performing the molecular design (Fig. 1d). De novo molecular design may be successful even if no molecules have been reported in the potential overlap of the ADs (upper right of Fig. 1d). However, there is no guarantee that the generative model will always succeed in designing molecules within the potential overlap (middle right of Fig. 1d). If the reliability levels are too strict (bottom right of Fig. 1d), the molecular design will be difficult, but whether it is actually impossible cannot be determined until the design process is executed.

To overcome this difficulty, we developed an optimization framework called DyRAMO (Dynamic Reliability Adjustment for Multi-objective Optimization), which performs multi-objective optimization while maintaining the reliability of multiple prediction models. DyRAMO was designed to adjust the appropriate reliability level for each objective, achieving a balance between high prediction reliability and predicted properties of the designed molecules. In DyRAMO, the reliability levels were explored through repeated molecular designs. To make this exploration efficient, Bayesian optimization (BO)[43] was integrated into this approach. To demonstrate the efficacy of DyRAMO, we designed epidermal growth factor receptor (EGFR) inhibitors while maintaining high reliability for three properties: inhibitory activity against EGFR, metabolic stability, and membrane permeability. Appropriate reliability levels were efficiently explored using BO, and promising molecules, including known inhibitors, were successfully designed. In addition, we face a situation in which the reliability needs to be varied for each property prediction in a practical multi-objective optimization. We reported that such a prioritization is also possible by adjusting the DyRAMO settings. These results demonstrate the effectiveness of the data-driven generative strategy using DyRAMO. The DyRAMO is available at https://github.com/ycu-iil/DyRAMO.

## Results
### Construction of DyRAMO
Figure 2 shows the DyRAMO framework for achieving multi-objective optimization while maintaining the prediction reliability as much as possible. Reliability levels are explored through an iteration of the following three steps: setting the reliability level of each property prediction, designing molecules, and evaluating the results of molecular design according to their predicted properties and their reliability levels (Fig. 2). Details of each step are described below.

In Step 1, a reliability level $\rho_i$ is set for each target property $i$, and the ADs of prediction models for the properties are defined based on the set reliability levels. To define AD in this study, we adopted the maximum value of Tanimoto similarity (MTS) for the training data, which is very simple but one of the most commonly used methods[9,44,45]. At a reliability level $\rho$, a molecule is included in the AD of a prediction model if the highest value of Tanimoto similarities between the molecule and the ones in the training data of the model exceeds $\rho$. The spread size of an AD varies with the reliability level $\rho$.

In Step 2, the molecules are designed using a generative model to enter the overlapping region of the ADs given in Step 1 and perform multi-objective optimization. In this study, we employed ChemTSv2[46], which has proven performance in various molecular designs ranging from photo-functional materials[11] to drug design[8,10,47], to generate molecules using a recurrent neural network (RNN)[48] and a Monte Carlo tree search (MCTS)[49]. Details of the ChemTSv2, and configuration setup for execution are described in the Methods section. The properties of the designed molecules were evaluated using prediction models based on supervised learning, and the details of the models used in this study are presented in the Methods section. The strategy for performing multi-objective optimization within the ADs is described in the next section. If the settings in Step 1 are not appropriate, the overlap of the prepared ADs may not be sufficiently wide, or the optimization of the predicted properties may fail.

In Step 3, the molecular design is evaluated for two aspects: whether the design is highly reliable and whether the designed molecules are sufficiently optimized for the properties, using the DSS (Degree of Simultaneous Satisfaction of prediction reliability and multiple property optimizations) score defined below:

$$DSS = \left(\prod_{i=1}^{n} Scaler_i(\rho_i)\right)^{\frac{1}{n}} \times Reward_{topX\%} \qquad (1)$$

Here, for each $i$th property, $Scaler_i$ is a scaling function that standardizes the reliability level $\rho_i$ to a value between 0 and 1 depending on its desirability. The scaling function parameters can also be adjusted when prioritization is desired among the properties to be optimized. $Reward_{topX\%}$ is the average of the top $X$ reward values for the designed molecules, which indicates how well the multi-objective optimization was achieved. In this study, the top 10% of the reward values for the designed molecules were considered. It should be noted that this framework is constructed to work well with other definitions of ADs and uncertainties of prediction models, as long as any parameter of reliability level is available.

Exploring all the combinations of reliability levels is time-consuming because it requires repetitive molecular design until an appropriate combination is found. BO was introduced to make this exploration efficient. This technique allows the efficient exploration of input variables to maximize (or minimize) a predefined objective function. Here, the search space is a combination of possible reliability levels of the target properties, and the objective variable is the DSS score.

## Strategy of molecular design incorporating ADs

In Step 2 of DyRAMO, molecular design aiming for multi-objective optimization with reliable predictions is performed using ChemTSv2. The design process in ChemTSv2 focuses on maximizing a predefined reward function. To optimize multiple predicted properties within the ADs of multiple prediction models, we define the reward as follows:

$$Reward = \begin{cases} \left(\prod_{i=1}^{n} v_i^{w_i}\right)^{\frac{1}{\sum_{i=1}^{n} w_i}} & \text{if } s_i \geq \rho_i \text{ for all } i = 1, 2, \ldots, n \\ 0 & \text{otherwise} \end{cases} \qquad (2)$$

where $s_i$ represents the MTS between the designed molecule and the training data for each prediction model for $i$th property, $\rho_i$ is a reliability level, $v_i$ is the desirability of the predicted property, $w_i$ is the weight, and $n$ is the number of properties to be optimized. As a molecular descriptor for computing Tanimoto similarity, Morgan fingerprint[50,51], whose radius and dimension were 2 and 2048, respectively, was applied. To calculate the desirability of the predicted properties, Gaussian scaling functions were defined for each property according to Brown et al.[52] (Fig. S2). Weight $w$ is allocated to each predicted property for prioritization during optimization. When calculating the reward function, the evaluation of the generated molecules was changed depending on whether the designed molecule was within the ADs of the property prediction models. If the MTS between the designed molecule and training data exceeds the reliability level defined for all target property predictions, the designed molecule is judged to be within the ADs. In this case, as a reward, the Dscore[53], which is the weighted geometric mean of the desirability of each property and is often used in the multi-objective optimization of molecular properties[5,10,54], is calculated. The Dscore is designed to optimize multiple objectives in a balanced manner as the score improves, without neglecting any specific objective. This balanced approach is particularly valuable in drug discovery, where achieving a reasonable degree of optimization for all objectives is critical. If the MTS is below the reliability level for any objective, it is judged to be outside the ADs, and the reward value is set to 0. By designing the

reward function in this manner, we aim to achieve multi-objective optimization of the molecular predicted properties within the ADs of multiple prediction models.

## Multi-objective optimization avoiding reward hacking in drug design

To verify the performance of DyRAMO, we conducted molecular design in the context of drug design. As a drug target, EGFR, which is targeted in the development of anticancer drugs[55], was selected. Drug design requires the optimization of pharmacokinetic properties in addition to activity against drug targets, as these properties are essential for drug efficacy and safety[56]. Here, we considered two pharmacokinetic properties and addressed three: inhibitory activity against EGFR, metabolic stability, and membrane permeability. In adjusting the reliability levels, each property can be weighted in the DSS score. The reliability level of each property prediction was standardized by Scaler in the calculation of the DSS score. The Scaler has an adjustable parameter $\sigma$ (Fig. S1), which is used to weight each property in adjusting reliability levels. Reducing the $\sigma$ makes the Scaler stricter and more sensitive to fluctuations in reliability levels. The $\sigma$ used in this study was three patterns: 0.15 (i) for tight, 0.25 (ii) for normal, and 0.35 (iii) for loose. The $\sigma$ for each property here was set to (i) for inhibitory activity, and (ii) for metabolic stability and membrane permeability. Details of the development of prediction models for these properties, the molecular design using ChemTSv2, and the search for appropriate reliability levels using DyRAMO are described in the Methods section. To verify the effect of the molecular design on the prediction reliability, the same molecular design without considering the reliability was performed. Furthermore, to consider that desirable molecules exist in the training data of the property prediction models, we removed the approved EGFR inhibitors from the training data and then performed molecular design using this modified dataset.

Figure 3 shows the results of the molecular designs with reliability levels adjusted by DyRAMO (Fig. 3a, c) and without considering the prediction reliability (Fig. 3b, d). The optimized reliability levels were 0.66 for inhibitory activity, 0.55 for metabolic stability, and 0.43 for membrane permeability, and the exploration process of the reliability levels is discussed in the next paragraph, and Fig. 4. Figure 3a shows the optimization process for the predicted properties and reliability of the molecules designed using ChemTSv2 with the adjusted reliability levels. The MTS between a designed molecule and the training data for each property was evaluated as the reliability of the designed molecule. The series in Fig. 3a and b show the moving averages of each of the 200 designed molecules. As the design progressed, the predicted values of the three properties, particularly the inhibitory activity, increased. The reliabilities of the designed molecules exceeded the set reliability levels (dotted lines) during molecular design, indicating that they were included in the adjusted AD for each model. Figures 3c and S3 show examples of the designed molecules along with the predicted property values and MTSs of the training data. These molecules were predicted to have an inhibitory activity against EGFR ($pIC_{50}$) of over 8, with the other two predicted properties also generally well optimized. From the viewpoint of drug design, these molecules have a quinazoline substructure (highlighted in Fig. 3c), a characteristic substructure of known EGFR inhibitors[57], suggesting that the essential feature of EGFR inhibitors is captured during the molecular design process. However, as shown in Fig. 3b and d, multi-objective optimization without considering prediction reliability resulted in reward hacking. Although the predicted properties were optimized, the designed molecules differed significantly from the training data. The designed molecules shown in Fig. 3d are far from known EGFR inhibitors. These results indicate that with the adjustment of the reliability levels by DyRAMO, a molecular design can be effectively achieved.

In addition, even when approved drugs were removed from the training data of the property prediction models, DyRAMO designed

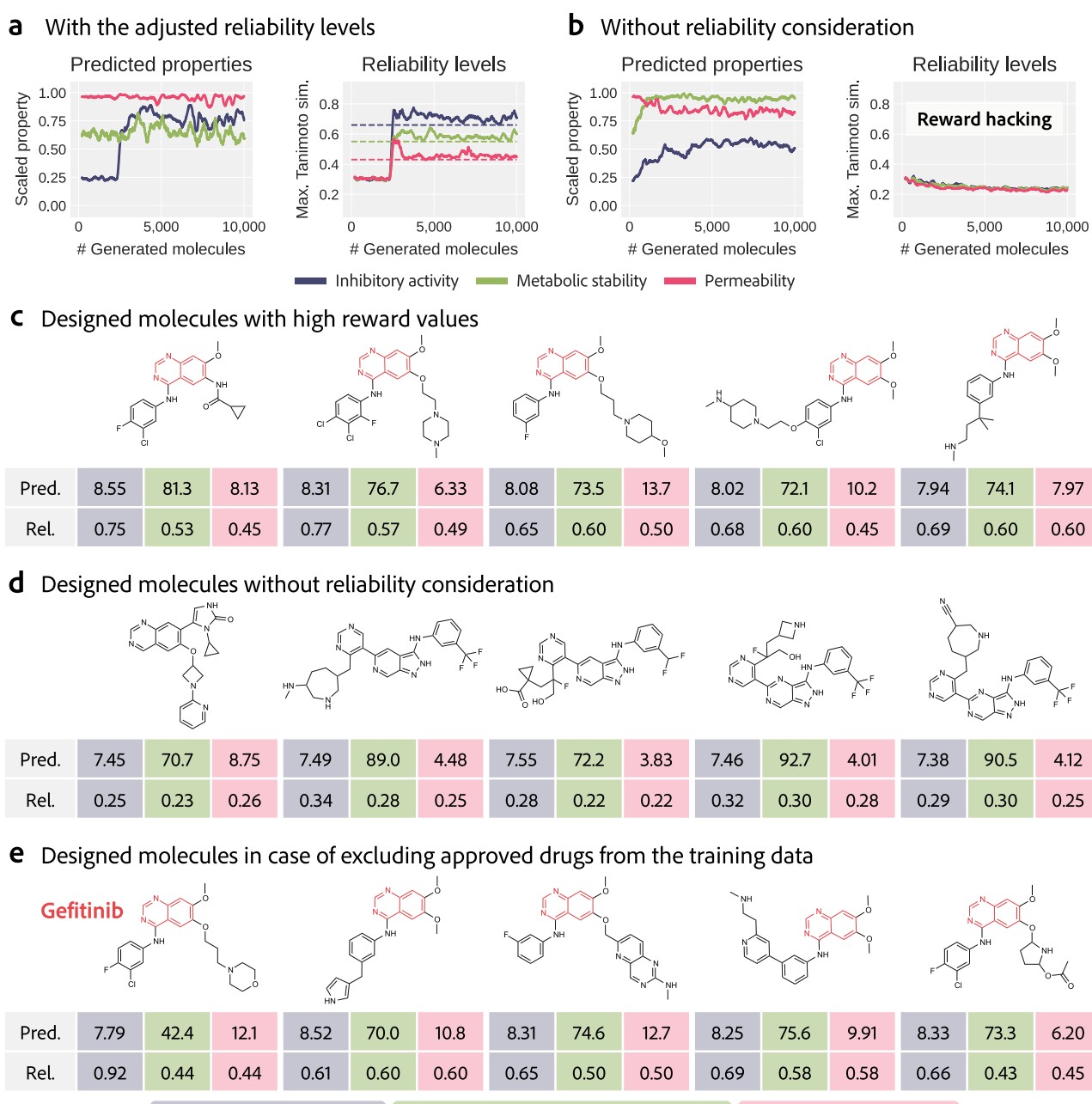

**Fig. 3 | Result of the molecular designs using DyRAMO.** Result of the molecular designs with adjusted reliability levels by DyRAMO (Dynamic Reliability Adjustment for Multi-objective Optimization) (**a**, **c**, **e**) and without considering the prediction reliability (**b**, **d**). **a**, **b**, The left panel shows the evolution of the predicted properties, with the predicted values scaled from zero to one. The right panel shows the evolution of the maximum value of Tanimoto similarity (MTS) between the designed molecules and the training data of each property. The dotted line represents the set reliability levels. **c**–**e** Examples of designed molecules and their predicted properties (Pred.) and prediction reliability, i.e., the MTS with the training data, of each property (Rel.). **c** Examples of molecules designed by DyRAMO with well-optimized predicted properties. The shown molecules were selected according to the following procedure. Molecules that exceeded the set reliability levels were obtained from ten times molecular designs with high DSS (Degree of Simultaneous Satisfaction of prediction reliability and multiple property optimizations) scores. Molecules that passed two filters, the rule of five filter and the PubChem filter, were extracted. The rule of five filter is a filter based on Lipinski's rule of five[80], and the PubChem filter[8] is a filter based on the frequency of occurrence of molecular patterns in the PubChem database. Subsequently, k-means

clustering (k=20) was conducted, and the molecules with the highest reward from each cluster were selected. Other designed examples are shown in Fig. S3. The highlighted areas of molecules represent the quinazoline substructure, a characteristic substructure of known epidermal growth factor receptor (EGFR) inhibitors. Inhibitory activity against EGFR (negative logarithm of the half-maximal inhibitory concentration: pIC50), metabolic stability (remaining percentage in one hour), and membrane permeability (µcm s⁻¹) are colored in blue, green, and red, respectively. Predictions are obtained from a single model run. **d** Examples of molecules designed without reliability consideration. The molecular designs for 10,000 molecules were conducted three times, and designed molecules that passed both the rule of five filter and the PubChem filter were extracted. K-means clustering was performed on these extracted molecules, and the molecule with the largest reward in each cluster was selected. Five molecules are shown here, and the others are shown in Fig. S4. **e** Designed molecules in the case where approved drugs were excluded from the training data of the property prediction models. Clustering was conducted to select molecules in the same manner as in (**c**). DyRAMO reproduced gefitinib, one of the approved drugs against EGFR (the left end of **e**). Five molecules are shown here, with the remaining molecules presented in Fig. S5.

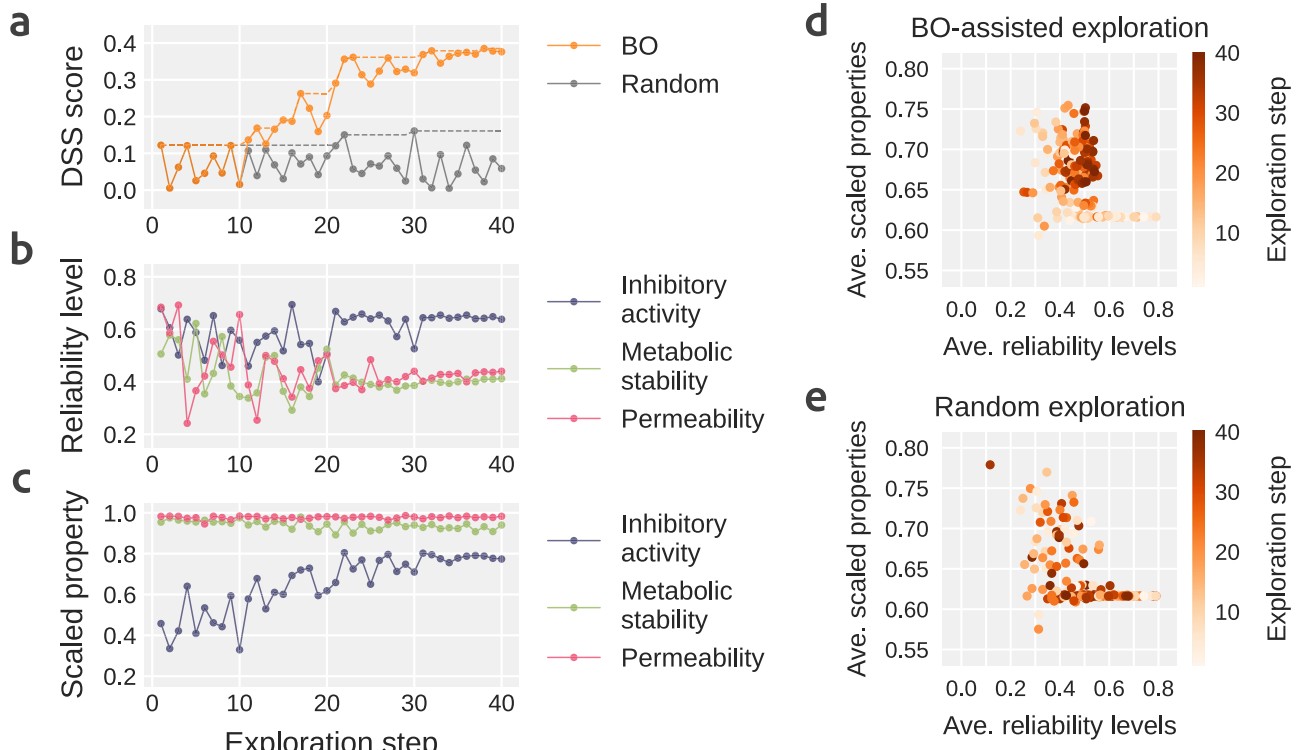

**Fig. 4 | The processes of adjusting reliability levels by Bayesian optimization (BO) and random exploration in DyRAMO (Dynamic Reliability Adjustment for Multi-objective Optimization).** Panels **a**–**c** show the evolution of the DSS (Degree of Simultaneous Satisfaction of prediction reliability and multiple property optimizations) score, reliability levels, and scaled predicted properties of designed molecules, respectively. The dashed lines in (**a**) trace the maximum DSS score achieved up to each exploration step. The search processes by BO (**d**) and random exploration (**e**) in the space of averaged reliability levels and the average scaled predicted properties. Each exploration step is color-coded, white at the beginning of the exploration and changing to orange as the exploration progresses. In the BO-assisted exploration, the exploration proceeded to optimize both the averaged reliability levels and scaled properties.

molecules with desirable predicted properties. Among the generated molecules, gefitinib, one of the approved drugs that had been removed from the training data of the property prediction models, was included (Fig. 3e). The number of molecules satisfying the reliability thresholds set by DyRAMO (i.e., classified as inside the ADs) was counted for both the newly generated molecules and the training molecules of the property prediction models. This analysis revealed that DyRAMO generated a sufficient number of novel molecules (Fig. S6, Table S1). Furthermore, examining the distribution of these molecules in chemical space demonstrated that the generated molecules not only overlapped with the space occupied by training molecules with desirable predicted properties but also extended into previously unexplored space (Fig. S7). These results highlight DyRAMO's capability to design novel and promising molecules with desirable predicted properties.

The exploration process for the optimal reliability levels using DyRAMO is shown in Fig. 4. For comparison, the results of randomly adjusting the reliability levels are also presented. Fig. 4a shows the evolution of the DSS score during BO-based and random exploration. In BO-based exploration, the DSS score increased as exploration progressed, eventually reaching approximately 0.38. In contrast, during random exploration, the DSS score fluctuated between 0.0 and 0.2 throughout. This demonstrates that the BO efficiently identified a combination of appropriate reliability levels that yielded a high DSS score. Figure 4b and c represent the evolution of the explored reliability levels and the scaled predicted properties of the designed molecules, respectively. Each scaled predicted property in each exploration step is calculated by extracting the designed molecules with the top 10% average of the three scaled properties from all

designed molecules, and then averaging each scaled property for these extracted molecules. The explored reliability levels stabilized late in the exploration process and converged at approximately 0.60 for inhibitory activity against EGFR, and 0.40 for metabolic stability and membrane permeability, respectively. The predicted properties were optimized as exploration progressed. Figure 4d illustrates how DyRAMO balances the desirability of the reliability levels and predicted properties. At the beginning of the exploration, either the average reliabilities or scaled properties are high, or both are low. However, as the exploration progresses, it can be seen that the exploration moves toward the upper right in an attempt to optimize both the averaged reliabilities and scaled properties. Figure 4e shows that it is difficult to conduct efficient random exploration. These results demonstrated the efficiency of BO-based exploration in DyRAMO for designing molecules while avoiding reward hacking.

## Prioritizing properties in reliability adjustment

In practical multi-objective optimization situations, the relative importance of each property may differ, and it is desirable to have high prediction reliability for properties of high importance, whereas for properties of lesser importance, it may be acceptable to compromise reliability. Therefore, we tested the feasibility of adjusting the reliability level of each property prediction by weighting it using the DyRAMO. This investigation was conducted under four scenarios: 1. without prioritization (no-priority) and 2. with the highest priority given to inhibitory activity against EGFR (inhibitory activity prioritized); 3. metabolic stability (metabolic stability prioritized); and 4. membrane permeability (permeability prioritized). The weighting was provided by the value of the $\sigma$ of the Scaler in the DSS score. The $\sigma$ for

each property was set to (i) for properties given the highest priority and (iii) for others. In pattern 1 (no-priority), (iii) was set for all properties. Except for the weighting, the reliability levels were explored under the same conditions as those in the exploration conducted by DyRAMO in Section Multi-objective optimization avoiding reward hacking in drug design.

Figure 5a shows that under the four scenarios with different reliability priorities, the molecules were optimized as intended. The vertical axis in Fig. 5a shows the reliability levels explored using the DyRAMO. In pattern 2 (inhibitory activity prioritized), the average of reliability levels for the inhibitory activity was higher than that in the no-priority pattern, approximately 0.63 and 0.53, respectively. Similar increases in the explored reliability levels were observed for patterns 3 (metabolic stability prioritized) and 4 (permeability prioritized). Figure 5b–d shows examples of molecules designed with patterns 3, 4, and 5. The similarity between the designed and training molecules for the prediction model was relatively high for the prioritized properties. In particular, when priority was given to metabolic stability or membrane permeability (patterns 3 and 4), whose optimized reliability levels were relatively low in pattern 1, slightly different molecules were designed, as shown in Fig. 5c and d. These results indicate that DyRAMO can adjust the reliability, reflecting priorities among the target properties.

## Molecular design in a situation where the overlap of ADs is not expected

In the context of multi-objective optimization using prediction models, it is assumed that the training molecules of each prediction model do not overlap or are not similar. In the molecular design examined in Section Multi-objective optimization avoiding reward hacking in drug design, several molecules, spanned multiple training datasets, leading to a situation in which the overlap of the ADs for each property was relatively large. Therefore, molecules likely to span multiple ADs were removed from the training data of the prediction model. Specifically, for data with MTS greater than 0.5, across the three datasets, a total of 777 molecules were removed (Fig. 6a). The prediction models were reconstructed using the partially removed dataset, and the reliability levels were explored using DyRAMO under the same conditions described in Section Multi-objective optimization avoiding reward hacking in drug design. ChemTSv2 with adjusted reliability levels successfully reproduced known EGFR inhibitors that were excluded from the training dataset (Fig. 6b). These molecules have high experimental $pIC_{50}$ values against EGFR (approximately 7-8), with generally optimized predicted metabolic stability and membrane permeability. This suggests that DyRAMO has the potential to identify drug candidates even when the molecules in each training dataset are not similar.

## Discussions

Reward hacking is a common phenomenon and an unavoidable issue as long as designing a perfect reward before optimization is difficult[23]. As has been pointed out in the past[21], reward hacking, in which multi-objective optimization seems to succeed, but the molecule is actually unpromising, occurred in this study, as shown in Fig. 3b, d. DyRAMO was designed with the intention of generating molecules within an overlapping reliable region by introducing the idea of adjustable AD for each property to avoid such reward hacking. As shown in Fig. 3, by appropriately adjusting the reliability levels, promising molecules, including known inhibitors, were designed by appropriately adjusting their reliability levels. In addition, because the exploration of an appropriate combination of reliability levels can be daunting, we formulated the reliability level exploration as a black-box optimization of a function that computes the DSS score that balances both the reliability and multi-objective optimization of the predicted properties. This allowed us to introduce an efficient exploration method using BO,

and the optimal reliability levels were found quickly, as shown in Fig. 4. Furthermore, as described in Section Prioritizing properties in reliability adjustment, properties can be prioritized by assigning appropriate weights to the DSS score. This study introduces a solution to the problem of multi-objective optimization using generative models, which is likely to become even more important in the future.

Although DyRAMO is a general framework for avoiding reward hacking in multi-objective optimization, there is still room for improvement, even when focusing on molecular design. First, it is necessary to investigate how to establish an appropriate definition of the reliability or confidence of a prediction model. Although this study adopted a simple and basic method of defining AD, MTS from the training data, various other definitions of AD have been proposed. For instance, in addition to distance-based methods, such as Tanimoto similarity, range-based and probability density distribution-based methods[58]. The molecular representations for these calculations include fingerprints, property-based vector representations, and graph representations[59]. The optimal approach for assessing prediction reliability differs for each dataset[58], making the optimization of reliability indicators crucial. DyRAMO is applicable to methods for defining ADs other than MTS and has also been confirmed to work effectively when using a 3D-based similarity metric to define ADs (Figs. S10, S11). Determining which AD indicators are most suitable for molecular design remains an issue that should be further investigated in the future. In addition, activity cliffs (ACs), a phenomenon where structurally similar molecules exhibit significantly different activities[60], require particular caution, especially defining ADs based on structure similarity-based metrics, such as MTS. In situations where ACs occur, the assumption that structural similarity correlates with higher prediction reliability may collapse, and the defined AD may not work as intended. A tentative approach to address this issue is to include occurrence of ACs as part of the criteria for defining ADs. Significant efforts have been made to predict whether a given pair of molecules may exhibit ACs[61–66], and these methods could potentially be utilized to address the issue. A simple approach to handling ACs using these methods would be to classify molecule pairs predicted to cause ACs as outside ADs. Although it is currently challenging to completely avoid prediction errors caused by ACs using machine learning-based AC detection methods[67], addressing these issues remains an important area for future investigation.

In addition, the design of the reward, that is, the evaluation function for molecular design and the DSS score, may also have areas for advancement. Careful setting of the evaluation function in molecular design is required because it directly affects the molecule to be designed; therefore, a suitable design for DyRAMO should be explored. Although multi-objective optimization was formulated using the Dscore[53] in this study, this serves merely as one example, other optimization strategies using weighted sums[7,54] or Pareto solutions[7,68,69] may also be promising. In this study, most of the reliability levels were low when the number of target property variables was high, as shown in Fig. S12 and Table S2. This reflects the difficulty of the problem itself, but it could be improved by refining the molecular design model, careful setting of the molecular design, and modifying the DSS scores.

Finally, to overcome the limitations of molecular designs that depend on training data, as shown in this study, the use of methods that do not rely on currently available data is essential. As specific strategies, adding new data or predicting properties through molecular simulations based on physicochemical backgrounds can be considered. Although adding new data is costly, it is expected to expand the AD, thereby enabling reliable molecular design in a broader chemical space. Incorporating the estimation of physical properties by quantum chemistry, all-atom, coarse-grained, or docking simulations into molecular design is also a promising strategy. These methods are based on physicochemical principles and thus could mitigate risks

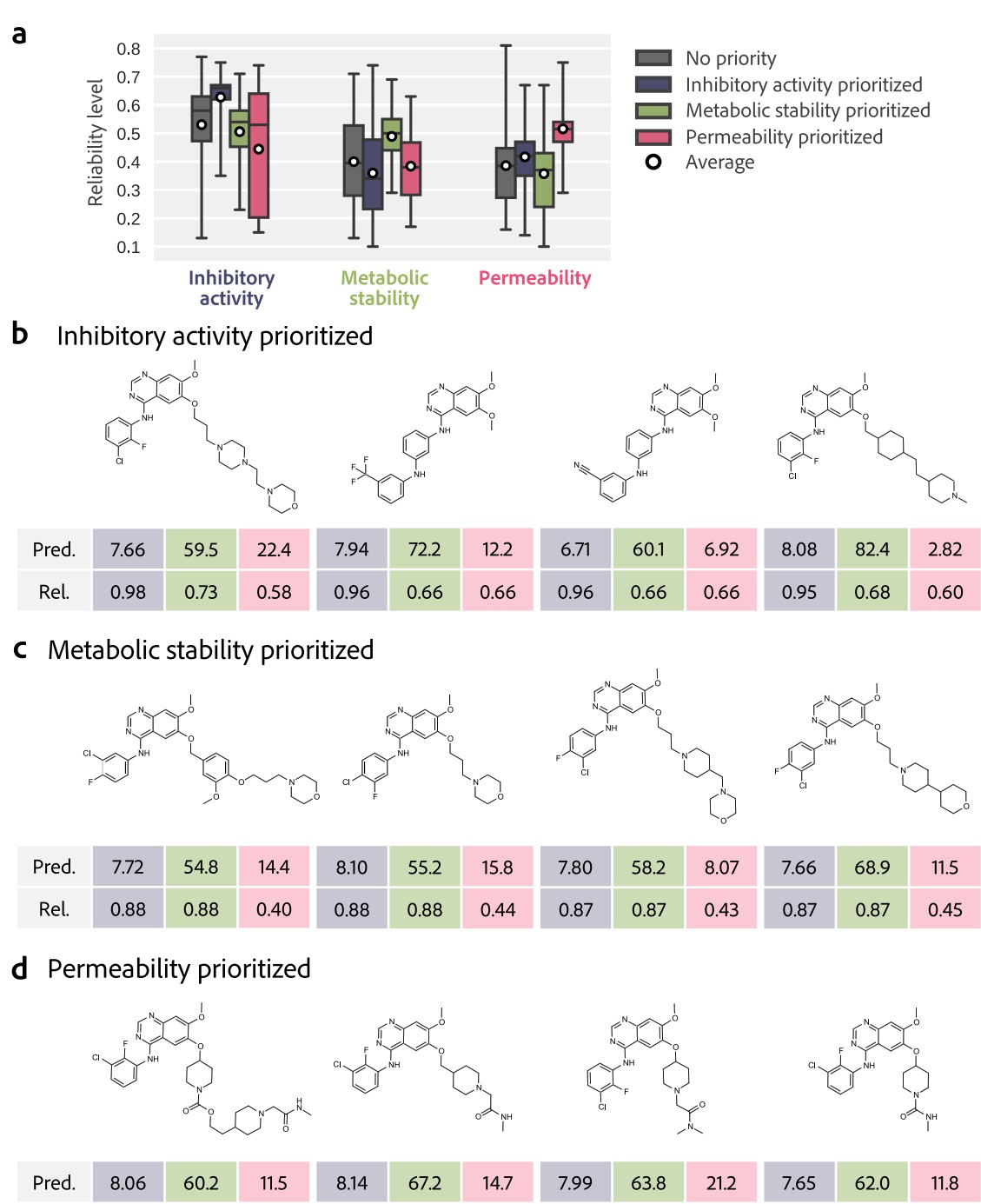

**Fig. 5 | Effects of prioritization on adjustment of reliability levels. a** Reliability levels explored in each prioritization pattern. From left to right, the explored reliability levels for inhibitory activity against epidermal growth factor receptor (EGFR), metabolic stability, and permeability are shown. The box-and-whisker plots represent the distributions of reliability levels under different prioritization patterns: no priority (gray), inhibitory activity prioritized (blue), metabolic stability prioritized (green), and permeability prioritized (red). Each box indicates the interquartile range, with the horizontal line representing the median of explored reliability levels. The whiskers show the range from 5th to 95th percentiles of the explored reliability levels. The circles indicate the average of explored reliability levels for each prioritization pattern. Each plot is based on 150 samples, generated through DyRAMO (Dynamic Reliability Adjustment for Multi-objective Optimization). DyRAMO performed 30 exploration steps with Bayesian optimization per run, repeated across 5 runs, resulting in a total of 150 samples for each prioritization pattern. Examples of molecules designed in the patterns prioritize inhibitory activity (**b**), metabolic stability (**c**), and permeability (**d**). Novel molecules with a high maximum value of Tanimoto similarity (MTS) to the training data of the prioritized property are shown. The values under each molecule present the predicted property (Pred.) and prediction reliability (Rel.), i.e., the MTS with the training data for each property. Inhibitory activity against EGFR (negative logarithm of the half-maximal inhibitory concentration: $pIC_{50}$), metabolic stability (remaining percentage in one hour), and membrane permeability ($\mu cm\,s^{-1}$) are colored in blue, green, and red, respectively.

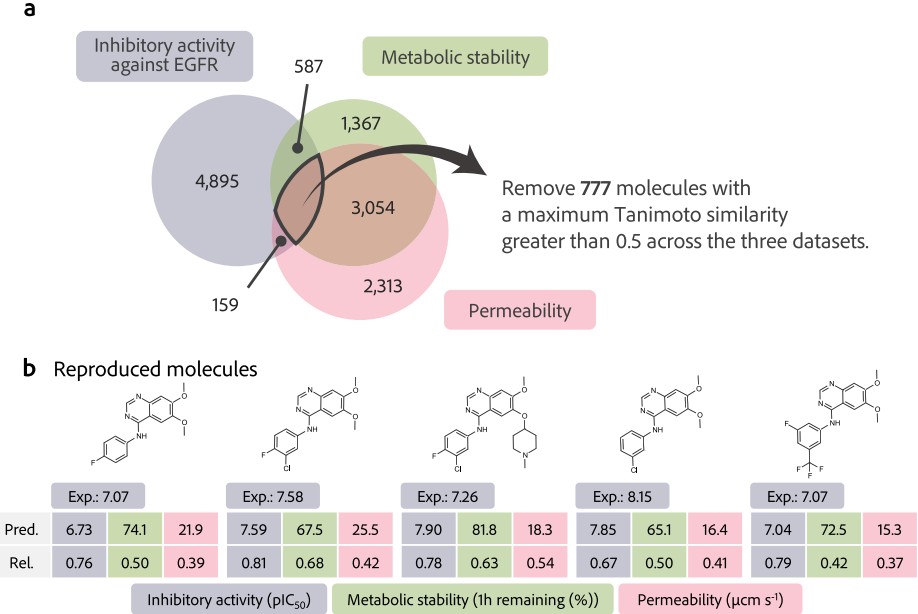

**Fig. 6 | Removal of similar molecules across datasets and rediscovered examples in the removed molecules. a** 777 molecules that were similar between datasets, specifically those with a maximum Tanimoto similarity (MTS) greater than 0.5, were removed. The numbers noted in the Venn diagram indicate the number of data. **b** Examples of molecules reproduced as a result of molecule design by DyRAMO (Dynamic Reliability Adjustment for Multi-objective Optimization). Experimental inhibitory activity values (Exp.) of the molecules are noted at the top of each table. Predicted properties (Pred.) and prediction reliability, i.e., the MTS with the training data, of each property (Rel.) of the molecules are presented in the tables. Inhibitory activity against epidermal growth factor receptor (EGFR) (negative logarithm of the half-maximal inhibitory concentration: $pIC_{50}$), metabolic stability (remaining percentage in one hour), and membrane permeability ($\mu cm\,s^{-1}$) are colored in blue, green, and red, respectively. Other reproduced molecules and designed molecules with high reward values are shown in Figs. S8 and S9, respectively.

associated with prediction reliability and activity cliffs, which are inherent in data-driven property estimation. Indeed, experimental validation of functional molecules designed by quantum chemistry calculations has been reported[18]. While it is challenging to simulate all properties, including permeability and pharmacokinetics, as addressed in this study, utilizing the available simulation techniques to verify and design molecules with low reliability will be necessary in the future.

## Methods
### Molecule generator
As a molecule generator, we employed ChemTSv2[46], which has been applied in materials[11,13,18,19] and drug design[8,10,47] and has been experimentally validated to have the ability to design desired molecules. ChemTSv2 combines two main components: molecular design with an RNN and chemical space exploration via MCTS. In MCTS, each node represents an atom in the molecule's Simplified Molecular Input Line Entry System (SMILES)[70] string, exploring molecules by extending nodes to build a search tree. The exploration process using MCTS involves four steps: selection, expansion, simulation, and backpropagation. In the selection step, a leaf node is selected based on the upper confidence bound (UCB) score. The UCB score balances exploration and exploitation by adjusting the parameter $C$. For instance, a high $C$ value (e.g., 1.0) emphasizes exploitation, focusing on unvisited nodes, whereas a low value (e.g., 0.1) encourages the exploration of promising nodes. In the expansion step, the selected node is expanded based on the RNN. In the simulation step, a molecule is generated by completing the SMILES part with the RNN, and is evaluated using a reward function. Finally, backpropagation updates the UCB score of the evaluated node based on the reward function. ChemTSv2 iterates these four steps to design molecules that maximize the value of the reward function. The RNN model was trained using 224,153 molecules from the ChEMBL database. These molecules were represented as randomized SMILES[71] to explore a wide range of chemical space.

### Construction and evaluation methods of the property prediction models
The construction and evaluation methods of the prediction models used to evaluate the designed molecules are described below. In this study, we tried two patterns for several target properties: 3 properties and 13 properties. When the 3 properties pattern, as target properties, the following three properties were selected: inhibitory activity against EGFR, metabolic stability, and membrane permeability. When 13 properties pattern, an additional ten properties were incorporated, including inhibitory activity against eight off-target proteins from tyrosine kinases and two other properties. Eight tyrosine kinase proteins were selected as described by Yoshizawa et al.[10]: receptor protein-tyrosine kinase erbB-2, Abelson tyrosine-protein kinase, proto-oncogene tyrosine-protein kinase, lymphocyte-specific tyrosine-protein kinase, platelet-derived growth factor receptor beta, vascular endothelial growth factor receptor 2, fibroblast growth factor receptor 1, and ephrin type-B receptor 4. For the other two properties, inhibitory activity against the human ether-a-go-go-related gene channel (hERG) and solubility were selected. The training data for these models were obtained from ChEMBL28[72] for inhibitory activity against tyrosine kinases, metabolic stability, and membrane permeability; from Sato et al.[73] for hERG inhibition; and from Tayyebi et al.[74] for solubility. To prepare the input features for the prediction models, Morgan fingerprints with radii and dimensions of 2 and 2048, respectively, were calculated using RDKit software[75]. As a prediction algorithm, light gradient boosting machine[76] was applied, and the hyperparameters were optimized using Optuna software[77]. The prediction performance of each model was evaluated using 5-fold cross-validation (Fig. S13). Finally, for each property, prediction models were developed using the entire dataset for the molecular design.

**Bayesian optimization-assisted adjustment of reliability levels**

The goal of DyRAMO is to optimize the DSS score to establish an appropriate reliability level for defining AD while iteratively designing molecules. The search spaces, consisting of combinations of reliability level value sets for each target property prediction, are defined as follows: The units of the reliability level candidates are adjusted based on the number of target properties. For the 3 properties case, we generated 80 options per property, spanning from 0.1 to 0.9 by 0.01 increments, yielding $80^3$ threshold combinations for exploration. For the 13 properties case, we prepared three reliability level candidates per property: 0.3, 0.4, and 0.5, resulting in $3^{13}$ possible combinations. DyRAMO uses BO to efficiently explore the appropriate combinations of reliability levels that result in higher DSS values. For the acquisition function in the BO, the expected improvement[78] was used in all computational scenarios. The computation of BO, a Python library, PHYSBO[79], was used. After 10 random explorations were conducted for initialization, 30 explorations were conducted using BO. This total of 40 explorations is called an episode. Five episodes were run for each set of computational scenarios to account for the randomness in the exploration processes.

## Data availability

Property prediction models and their training data are available on GitHub at https://github.com/ycu-iil/prediction_model_collection. Source data are provided with this paper.

## Code availability

The Python code used to implement DyRAMO is available on GitHub at https://github.com/ycu-iil/DyRAMO and Zenodo at https://doi.org/10.5281/zenodo.14731974. DyRAMO is available under the MIT License. Methods for reproducing the results are described in the following: https://github.com/ycu-iil/DyRAMO/blob/main/doc/reproduction_instruction.md.

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

## Acknowledgements

This research was conducted in "Development of a Next-generation Drug Discovery AI through Industry-Academia Collaboration (DAIIA)" under grant no.JP23nk0101111 (T.H. and K.T.) and a Research Support Project for Life Science and Drug Discovery under grant no.J-P22ama121023 (K.T.) from Japan Agency for Medical Research and Development (AMED). This work was also supported by the Ministry of Education, Culture, Sports, Science and Technology (MEXT) under the grants: Simulation- and AI-driven next-generation medicine and drug discovery based on "Fugaku" (Grant Number: JPMXP1020230120, K.T.), feasibility studies for the next-generation computing infrastructure, Data Creation and Utilization Type Material Research and Development Project (Grant Number: JPMXP1122683430, K.T.), JST FOREST Program (Grant Number: JPMJFR232U, K.T.), RIKEN Junior Research Associate Program (T.Y.), and Chugai Foundation for Innovative Drug Discovery Science (C-FINDs) (T.Y.).

## Author contributions

Tatsuya Yoshizawa: conceptualization (lead); methodology (lead); software (lead); writing—original draft (lead); writing—review and editing (lead). Shoichi Ishida: conceptualization (support); methodology (lead); software (support); writing—review and editing (lead). Tomohiro Sato: methodology (support); software (support), writing—review and editing (support). Masateru Ohta: methodology (support); software (support), writing—review and editing (support). Teruki Honma: supervision (support); funding acquisition (lead); project administration (lead); methodology (support); writing—review and editing (support). Kei Terayama: supervision (lead); funding acquisition (lead); methodology (lead); project administration (lead); conceptualization (lead); software (support); writing—review and editing (lead).

## Competing interests

The authors declare no competing interests.
