## [Transparent Peer Review file · Nature Communications]

Avoiding Reward Hacking in Multi-Objective Molecular Design: A Data-Driven Generative Strategy with Reliable Design Framework

Corresponding Author: Dr Kei Terayama

This manuscript has been previously reviewed at another journal that is not operating a transparent peer review scheme. The manuscript was considered suitable for publication without further review at Nature Communications.

Version 0:

Reviewer comments:

Reviewer #1

(Remarks to the Author)

The manuscript describes significant progress in the field of molecular design using data-driven generative modelling. The proposed reliable design framework addresses a key issue in multi-objective optimisation, in particular the challenge of rewarding hacking and the difficulty of predicting reliability in chemical space. The authors provide a compelling case study of anti-cancer drug design that demonstrates the effectiveness of the framework. Overall, this is an interesting study. But the study lacks methodological innovation as it is Nature Communications. I do not recommend articles for publication in Nature Communications.

Comments:

1. The method in this study is based on the integration of ChemTSv2, lacking methodological innovation.
2. The method section uses Tanimoto similarity, which is only a two-dimensional topological similarity method for molecular structures and is relatively simplistic. It would be worthwhile to consider evaluating three-dimensional similarities, such as molecular shape similarity.
3. Although many experimental evaluations have been conducted, I still feel that the novelty and impact of this paper are insufficient for publication in Nature Communications.

(Remarks on code availability)

Reviewer #2

(Remarks to the Author)

Molecular generation that is guided by surrogate models can often lead to the generation of compounds that score very well according to the surrogate model but are out-of-distribution. The generation of such compounds may be undesirable, as their predicted model scores are generally uncertain. In multi-objective molecular design, the challenge of generating in-distribution compounds is exacerbated, as score reliability must be maintained for all surrogate models. Yoshizawa et al address this challenge by proposing a generative strategy that aims to design compounds with promising and reliable scores, according to a Tanimoto reliability metric. The comparisons between the proposed method and an alternative without reliability consideration demonstrate the method's superiority in generating compounds with reasonable structures and scores that have a high reliability metric. The text is written clearly and the Figures neatly illustrate the proposed method and results. However, concerns remain regarding the treatment of training data as compounds "designed" by DyRAMO. If this concern and others described below are addressed, Nature Communications is an appropriate venue for publication of this manuscript.

Comments

- 1) Compounds in the training set for one or more surrogate models will naturally be favored by DyRAMO, as they will have

higher reliability levels for at least one model. This results in DyRAMO “designing” compounds in the training set(s) of one or more models (e.g., gefitinib). However, a simple virtual screen of the union of training sets for all models would yield gefitinib without the use of a generative model. Compounds in the training sets of any surrogate model should therefore not be included in the evaluation of DyRAMO. If DyRAMO is intended for use in generative design, which the introduction suggests, compounds that could be designed through a virtual screen should not be included in the results.

2) As a continuation of the previous comment, the authors may consider a virtual screen of the union of training sets as an additional baseline. Here, the DSS score would be estimated for every compound in the training set of at least one model, and DyRAMO’s designs would be the compounds with the highest scores. This would demonstrate DyRAMO’s ability to generate novel compounds (relative to the training sets) that still maintain high scores with high reliability.

3) The primary application of this work is multi-objective molecular design, and the approach to multi-objective optimization will necessarily impact the designs. Therefore, Equation 2, which defines the scalarization function used to combine multiple objectives, should be included in the main text. Additionally, can the authors please clarify why they chose the defined scalarization function instead of the more common weighted sum scalarization?

4) As mentioned in comment 1, one potential limitation of this approach is the preference for compounds that are similar to the training set compounds. This may limit DyRAMO’s ability to explore chemical spaces beyond the training sets. There is also the inherent assumption of DyRAMO that higher molecular similarity leads to higher prediction reliability. However, activity cliffs are common in structure-property relationships and may indicate that this assumption fails under certain conditions. Can the authors please address these limitations in the discussion, and perhaps propose future work that may mitigate them?

5) The authors occasionally fail to distinguish between true properties and predicted properties. This distinction is important in this manuscript, as property predictions are being optimized in an effort to arrive at compounds with decent experimental properties. In line 112, for example, “each property” should be replaced with “each property prediction”.

(Remarks on code availability)

The Github repository appears to be well-organized and contains clear directions for running DyRAMO. DyRAMO’s code was not tested as part of this review.

Version 1:

Reviewer comments:

Reviewer #1

(Remarks to the Author)

No further comment. Thank you for your kind responses, and I am happy that we can now have almost the same understanding of your research.

(Remarks on code availability)

Reviewer #2

(Remarks to the Author)

The authors have provided thorough responses to the comments and updated their manuscript with additional experiments to support their claims. The removal of training set compounds from the results, the addition of a new experiment after removing approved inhibitors, and the report of the number of “overlapping” designs are particularly appreciated. This manuscript in its current state appears to be suitable for publication in Nature Communications.

(Remarks on code availability)

The Github repository appears to be well-organized and contains clear directions for running DyRAMO. DyRAMO’s code was not tested as part of this review.

Responses and disposition to the comments from the reviewers

- **Responses and disposition in general**

We thank the reviewers for their valuable comments which have helped in improving the quality of our manuscript. According to their comments, we conducted additional experiments and revised the original text and figures. Our point by point responses to reviewer comments are given below.

- **Item to item response and disposition to the comment of Reviewer 1:**

Comment:

The manuscript describes significant progress in the field of molecular design using data-driven generative modelling. The proposed reliable design framework addresses a key issue in multi-objective optimisation, in particular the challenge of rewarding hacking and the difficulty of predicting reliability in chemical space. The authors provide a compelling case study of anti-cancer drug design that demonstrates the effectiveness of the framework. Overall, this is an interesting study. But the study lacks methodological innovation as it is Nature Communications. I do not recommend articles for publication in Nature Communications.

Response and disposition:

We appreciate the reviewer's acknowledgment of the significance and progress presented in this study. While we recognize the reviewer's concern about the level of methodological innovation, we believe our approach demonstrates meaningful advancement in the practical multi-objective molecular design. Below, we address specific comments and detail the revisions made.

Comment 1:

The method in this study is based on the integration of ChemTSv2, lacking methodological innovation.

Response and disposition:

Thank you for pointing this out. The core novelty of our framework is introducing a reliability-aware approach to molecular design. Multi-objective molecular design using generative models has long been hampered by the issue of reward hacking, which results in the generation of unreliable molecules. Our framework addresses this challenge by introducing a solution to reward hacking, enabling practical and reliable multi-objective molecular design for the first time. The application of this framework is not limited to ChemTSv2, although we employed it in this study. We hope readers will find this contribution valuable.

Comment 2:

The method section uses Tanimoto similarity, which is only a two-dimensional topological similarity method for molecular structures and is relatively simplistic. It would be worthwhile to consider evaluating three-dimensional similarities, such as molecular shape similarity.

Response and disposition:

We appreciate the reviewer's suggestion regarding the use of 3D similarity. In response, we conducted an additional

experiment using a 3D similarity metric, as an alternative to the 2D fingerprint-based Tanimoto similarity for the criterion of applicability domains (ADs). As a 3D similarity metric, we employed the Tanimoto combo similarity, the sum of shape-based Tanimoto similarity and pharmacophore-based Tanimoto similarity (commonly referred to as color Tanimoto similarity). To calculate the Tanimoto combo similarity, we employed shapescreeen[1], an open source software for a 3D-based similarity scoring. The 3D molecular conformations that are required for this calculation were generated using the ETKDGV3 method[2] in RDKit. Apart from the criterion for ADs, conditions for molecular designs and Bayesian optimization remained identical to those described in Section 2.3.

The results in the following Figure 1 and 2 show that DyRAMO effectively designed desirable molecules even when using the 3D-based similarity metric as criterion for ADs. Figure 1 presents the results of molecular design under adjusted reliability levels using DyRAMO (Figure 1a, c), alongside the results without considering reliability (Figure 1b). Figures 1a and 1b show the evolutions of predicted properties (left panels) and reliability levels (right panels), where the reliability levels are represented as Tanimoto combo similarity values (which range from 0 to 2) divided by 2. The result indicate that the three properties improved as optimization progressed in both cases. When using DyRAMO (Figure 1a), the reliability levels of the designed molecules exceeded the thresholds set (dashed lines) during the molecular design process, which means designed molecules remained within the adjusted AD of each prediction model. In contrast, without considering reliability (Figure 1b), the reliability levels stayed low at around 0.4. Figure 1c shows some of the molecules designed by DyRAMO, which are predicted to have desirable properties at sufficient reliability levels.

Figure 2 illustrates the exploration process for reliability levels using DyRAMO. In this process, DyRAMO efficiently identified appropriate reliability levels, consistent with the results shown in Figure 4 of the Results section, where 2D Tanimoto similarity was used as criterion for ADs. Figure 2a shows the DSS score progression during BO-based exploration compared to random exploration. In the BO-based approach, DSS scores increased as optimization progressed, eventually exceeding 0.4. In contrast, DSS scores in the random exploration fluctuated between 0.0 and 0.3, demonstrating that BO efficiently identified suitable reliability level combinations that resulted in high DSS scores. Figures 2b and 2c present the transitions of the explored reliability levels and the scaled properties of the designed molecules, respectively. Reliability levels converged to approximately 0.70 for EGFR inhibitory activity and around 0.50 for metabolic stability and membrane permeability during the later stages of exploration. Predicted properties slightly improved over the course of the exploration. Figure 2d illustrates how DyRAMO balanced reliability levels with predicted properties. Early in the exploration, either the averaged reliability levels or the averaged scaled property values were high, but rarely both. As the exploration progressed, the exploration shifted toward the upper-right corner, indicating an improvement in balancing reliability and predicted properties.

The source code used for molecule generation with this 3D similarity evaluation is also available on GitHub (https://github.com/yucu-ii/DyRAMO/blob/main/doc/reproduction_instruction.md#results-in-supplementary-information-molecular-design-using-tanimoto-combo-as-a-criterion-for-defining-ads). The ability of DyRAMO to incorporate various AD criteria, including 3D-based methods, serves as a foundation for advancing practical molecular design strategies in the future.

We believe these additional validations provide valuable insights for readers and have included the results and descriptions in the Discussion and Conclusion sections, as well as in the Supplementary Information, as follows:

(lines 351–355; page 14) “DyRAMO is applicable to methods for defining ADs other than MTS and has also been confirmed to work effectively when using a 3D-based similarity metric to define ADs (Figs. S10, S11). Determining which AD indicators are most suitable for molecular design remains an issue that should be further investigated in the future.”

We also added Figure S10 (Figure 1 in the response) and Figure S11 (Figure 2 in the response) in Supporting Information.

References

1. Shapescreeen (Chemical Data Processing Toolkit): <https://cdpkit.org/v1.1.1/applications/shapescreeen.html> (accessed December 13, 2024)
2. Riniker, S., *et al.* Better informed distance geometry: using what we know to improve conformation generation. *J. Chem. Inf. Comp. Sci.* **55**, 2562–2574 (2015).

Figure 1. Results of the molecular designs with adjusted reliability levels by DyRAMO (a, c) and without considering prediction reliability (b) in case the Tanimoto combo was employed as a criterion for ADs. **a, b,** The left panels show the moving averages of the predicted properties, with predicted values scaled from 0 to 1. The right panels show the moving averages of prediction reliability, represented as the Tanimoto combo similarity values (ranging from 0 to 2) divided by 2 between the designed molecules and the training data of each property. These moving averages were calculated from each of the 200 designed molecules. The dotted lines indicate the set reliability levels. The right panel of (b) is identical to that in Figure 3b in the manuscript. **c,** Examples of designed molecules, their predicted properties (Pred.), and their prediction reliability (Rel.), represented as the Tanimoto combo similarity divided by 2 with respect to the training data for each property.

Figure 2. The processes of adjusting reliability levels by BO and random exploration in DyRAMO in case the Tanimoto combo was employed as a criterion for ADs. **a-c**, a, b, and c show the evolution of the DSS score, reliability levels, and scaled predicted properties of designed molecules, respectively. **d, e**, The search processes by BO (d) and random exploration (e) in the space of averaged reliability levels and the average scaled predicted properties. Each exploration step is color-coded, white at the beginning of the exploration and changing to orange as the exploration progresses.

Comment 3:

Although many experimental evaluations have been conducted, I still feel that the novelty and impact of this paper are insufficient for publication in Nature Communications.

Response and disposition:

We appreciate the reviewer's feedback. While the novelty may appear to increase only incrementally, we believe that integrating reliability into multi-objective molecular design offers practical significance. We hope the revised manuscript better conveys the impact of our work.

- **Item to item response and disposition to the comment of Reviewer 2:**

Comment:

Molecular generation that is guided by surrogate models can often lead to the generation of compounds that score very well according to the surrogate model but are out-of-distribution. The generation of such compounds may be undesirable, as their predicted model scores are generally uncertain. In multi-objective molecular design, the challenge of generating in-distribution compounds is exacerbated, as score reliability must be maintained for all surrogate models. Yoshizawa et al address this challenge by proposing a generative strategy that aims to design compounds with promising and reliable scores, according to a Tanimoto reliability metric. The comparisons between the proposed method and an alternative without reliability consideration demonstrate the method's superiority in generating compounds with reasonable structures and scores that have a high reliability metric. The text is written clearly and the Figures neatly illustrate the proposed method and results. However, concerns remain regarding the treatment of training data as compounds "designed" by DyRAMO. If this concern and others described below are addressed, Nature Communications is an appropriate venue for publication of this manuscript.

Response and disposition:

We thank the reviewer for recognizing the significance and progress of our proposed generative strategy. Regarding the concern about the treatment of training data, we have acknowledged its importance and conducted additional experiments and analyses to accurately evaluate DyRAMO's ability to design novel and promising molecules. We hope these results clarify the robustness of our approach and its potential for multi-objective molecular design.

Comment 1:

Compounds in the training set for one or more surrogate models will naturally be favored by DyRAMO, as they will have higher reliability levels for at least one model. This results in DyRAMO "designing" compounds in the training set(s) of one or more models (e.g., gefitinib). However, a simple virtual screen of the union of training sets for all models would yield gefitinib without the use of a generative model. Compounds in the training sets of any surrogate model should therefore not be included in the evaluation of DyRAMO. If DyRAMO is intended for use in generative design, which the introduction suggests, compounds that could be designed through a virtual screen should not be included in the results.

Response and disposition:

Thank you for insightful opinion on this and the next comment. First of all, in response to the reviewer's comment, we removed generated molecules that are included in the training data from the following figures: one molecule from Figure 3c, four molecules from Figure 5b, one molecule from Figure 5c, and three molecules from Figure 5d. Additionally, Figure 3d and Figure S3 presented generated molecules overlapping with the training data.

To address the reviewer's concerns, we performed a more rigorous evaluation. This evaluation was designed to clarify whether DyRAMO's molecule design surpasses a capability of "virtual screening." Specifically, to investigate whether promising molecules like gefitinib could be generated as "novel" molecules, we created a modified dataset by removing the approved EGFR inhibitors from the training sets. The surrogate models are then retrained using this dataset, and molecular design was performed with DyRAMO. The detailed computational conditions, other than the

used dataset, were identical to those described in Section 2.3. As a result, DyRAMO successfully generated novel molecules, not included in the training data, with desirable predicted properties including gefitinib, as shown in Figure 3.

This result has been added to the Result section of the manuscript as follows:

- (lines 211–213; page 8) “Furthermore, to consider that desirable molecules exist in the training data of the property prediction models, we removed the approved EGFR inhibitors from the training data and then performed molecular design using this modified dataset.”
- (lines 240–244; page 8) “In addition, even when approved drugs were removed from the training data of the property prediction models, DyRAMO designed molecules with desirable predicted properties. Among the generated molecules, gefitinib, one of the approved drugs that had been removed from the training data of the property prediction models, was included (Fig. 3e).”

We also added Figure 3e (upper part of Figure 3 in the response) in Results section and Figure S5 (lower part of Figure 3 in the response) in Supporting Information.

Further discussion regarding the “virtual screening” is provided in our response to the next comment.

Figure 3. Examples of molecules designed by DyRAMO using the modified dataset. Values of predicted properties (Pred.) and prediction reliability, i.e., the maximum Tanimoto similarity with the training data, of each property (Rel.) are shown in each generated molecule. Inhibitory activity against EGFR (pIC₅₀), metabolic stability (remaining percentage in one hour), and membrane permeability (μm/s) are colored in blue, green, and red, respectively. The molecule at the top left is gefitinib, one of the approved drugs removed from the training data.

Comment 2:

As a continuation of the previous comment, the authors may consider a virtual screen of the union of training sets as an additional baseline. Here, the DSS score would be estimated for every compound in the training set of at least one model, and DyRAMO's designs would be the compounds with the highest scores. This would demonstrate DyRAMO's ability to generate novel compounds (relative to the training sets) that still maintain high scores with high reliability.

Response and disposition:

Thank you for the suggestion about “virtual screening” based evaluation. Per the reviewer's comment, we evaluated overlap between molecules in the training sets and those generated by DyRAMO and their predicted properties. For this analysis, we used the results of molecular designs under two conditions: (1) the molecular design described in the previous response (resp. 1 condition), and (2) the molecular design described in the main text (main condition). First, to confirm that DyRAMO could generate novel molecules, we categorized the generated molecules and training molecules into three groups: generated molecules that are not included in the training set (“novel generated molecules”), molecules that are found only in the training set (“unique training molecules”), and molecules that overlap between the generated set and the training set (“overlapped molecules”). The following Figure 4 and Table 1 show the number of molecules in these categories, counted based on reward value thresholds. For the resp. 1 condition, DyRAMO generated a substantial number of novel molecules, with only a small overlap with the training set. Among the generated molecules, some were predicted to have promising properties with reward values exceeding 0.9 (Figure 4a, upper section of Table 1). Similarly, for the main condition, DyRAMO was able to generate novel and promising molecules (Figure 4b, lower section of Table 1).

Furthermore, we visualized the distribution of the generated molecules and the training set in the chemical space by applying t-SNE. Figures 5a and 5b show the chemical space distributions for the resp. 1 condition and the main condition, respectively. In both cases, the generated molecules occupied regions that were distinct from the training set, while some of the generated molecules were also distributed within the same chemical space as the training set, covering its distribution of molecules with desirable predicted properties. These results also indicate that DyRAMO successfully generated novel molecules with desirable properties, as illustrated by their distribution in unexplored regions of the chemical space.

These results have clarified DyRAMO's capabilities as a generative model thanks to the reviewer's comments. Therefore, we have included these results in the manuscript as follows:

- (lines 244–253; pages 8–9) “The number of molecules satisfying the reliability thresholds set by DyRAMO (i.e., classified as inside the ADs) was counted for both the newly generated molecules and the training molecules of the property prediction models. This analysis revealed that DyRAMO generated a sufficient number of novel molecules (Fig. S6, Table S1). Furthermore, examining the distribution of these molecules in chemical space demonstrated that the generated molecules not only overlapped with the space occupied by training molecules with desirable predicted properties but also extended into previously unexplored space (Fig. S7). These results highlight DyRAMO's capability to design novel and promising molecules with desirable predicted properties.” We also added Figure S6 (Figure 4a in the response), Table S1 (upper part of Table 1 in the response) and Figure S7 (Figure 5a in the response) in Supporting Information.

Figure 4. Comparison of molecule counts across reward value thresholds for generated molecules, training molecules, and overlapped molecules. (a) Results under the resp. 1 condition. (b) Results under the main condition. Bars represent the count of molecules in three categories: novel generated molecules (orange), unique training molecules (dark blue), and overlapped molecules (light blue). Molecule counts are categorized by reward value thresholds (0.6, 0.7, 0.8, and 0.9). Here, the counted generated molecules were selected from the top 10 DSS score designs across all designs and were filtered to include only those that satisfied the ADs specific to each exploration step during DyRAMO’s process. Similarly, the training set molecules included in the count were those that satisfied one or more of these same ADs, ensuring consistency in the comparison.

Table 1. Molecule counts of the three categories. The table summarizes the number of molecules under the resp. 1 condition and the main condition, categorized into novel generated molecules, overlapped molecules, and unique training molecules. These results correspond to the data presented in Figure 4.

Reward value		≥ 0.6	≥ 0.7	≥ 0.8	≥ 0.9
Resp. 1 condition	Novel generated molecules	23,477	21,065	13,638	2,230
	Overlapped molecules	39	33	20	2
	Unique training molecules	1,116	872	587	249
Main condition	Novel generated molecules	20,493	16,715	9,612	1,458
	Overlapped molecules	29	22	13	4
	Unique training molecules	514	412	270	97

Figure 5. Chemical space distributions for the generated and training molecules under the resp. 1 condition (a) and the main condition (b). The plots were generated by applying t-SNE to the distance matrix constructed from Morgan fingerprints and Tanimoto similarity. Orange points represent 10000 molecules with high reward values from novel generated molecules, while blue points represent unique training molecules and overlapped molecules. The color intensity corresponds to reward values, with darker shades indicating higher values.

Comment 3:

The primary application of this work is multi-objective molecular design, and the approach to multi-objective optimization will necessarily impact the designs. Therefore, Equation 2, which defines the scalarization function used to combine multiple objectives, should be included in the main text. Additionally, can the authors please clarify why they chose the defined scalarization function instead of the more common weighted sum scalarization?

Response and disposition:

According to the reviewer's comment, we have moved Equation 2 and its related descriptions to the main text (lines 163–190; page 7).

We also appreciate reviewer's raising the important point regarding the scalarization function. In our study, we employed the weighted product approach as it is considered effective for optimizing multiple objectives in a balanced manner as the score improves, without neglecting any specific objective. If one objective has a low score, the overall product will inevitably become small. To maximize the overall product, all target variables need to be improved in a balanced manner. Since this study focuses mainly on molecular design for drug discovery, where balanced optimization of all objectives is critical, we chose the weighted product approach. Dscore was used in previous studies on EGFR inhibitor design [1,2].

Nonetheless, we acknowledge that the weighted sum is also a common approach and promising alternative. To confirm this, we conducted additional experiments to evaluate DyRAMO's performance using the weighted sum instead of the weighted product. The results showed that DyRAMO's exploration worked effectively with the weighted sum as well, achieving similar performance to the weighted product (Figures 6–8). We chose to use the weighted product as a representative example, but the weighted sum is equally a viable and widely used approach for the reward function.

To ensure readers understand this, we added descriptions and references to the Results section and the Discussion and Conclusion sections highlighting these points. The specific additions are as follows:

- (lines 183–187; page 7) The Dscore is designed to optimize multiple objectives in a balanced manner as the score improves, without neglecting any specific objective. This balanced approach is particularly valuable in drug discovery, where achieving a reasonable degree of optimization for all objectives is critical.
- (lines 373–375; page 15; A highlighted part is inserted.) “Although multi-objective optimization was formulated using the Dscore in this study, **this serves merely as one example**, other optimization strategies using weighted sums[3,4] or Pareto solutions may also be promising.”

References

1. Winter, R. *et al.* Efficient multi-objective molecular optimization in a continuous latent space. *Chemical Science* **10**, 8016–8024 (2019).
2. Yoshizawa, T. *et al.* Selective inhibitor design for kinase homologs using multiobjective monte carlo tree search. *Journal of Chemical Information and Modeling* **62**, 5351–5360 (2022).
3. Blaschke, T. *et al.* Reinvent 2.0: An ai tool for de novo drug design. *Journal of Chemical Information and Modeling* **60**, 5918–5922 (2020).
4. Liu, X. *et al.* Drugex v2: de novo design of drug molecules by pareto-based multi-objective reinforcement learning in polypharmacology. *Journal of Cheminformatics* **13** (2021).

Figure 6. The processes of adjusting reliability levels by DyRAMO. a, b, and c show the evolution of the DSS score, reliability levels, and scaled predicted properties of designed molecules, respectively. This figure corresponds to Figure 4a–c in the main manuscript.

Figure 7. Result of the molecular designs with adjusted reliability levels. The left panel shows the moving averages of the predicted properties, with the predicted values scaled from zero to one. The right panel shows the moving averages of the maximum Tanimoto similarity between the designed molecules and the training data of each property. These moving averages were calculated from each of the 200 designed molecules. The dotted line represents the set reliability levels. This figure corresponds to Figure 3a in the main manuscript.

Figure 8. Examples of molecules designed by DyRAMO. Values of predicted properties (Pred.) and prediction reliability, i.e., the maximum Tanimoto similarity (MTS) with the training data, of each property (Rel.) are shown in each generated molecule. Inhibitory activity against EGFR (pIC₅₀), metabolic stability (remaining percentage in one hour), and membrane permeability (μm/s) are colored in blue, green, and red, respectively. The property values shown in the tables are predicted values. This figure corresponds to Figures 3c and S3 in the manuscript.

Comment 4:

As mentioned in comment 1, one potential limitation of this approach is the preference for compounds that are similar to the training set compounds. This may limit DyRAMO's ability to explore chemical spaces beyond the training sets. There is also the inherent assumption of DyRAMO that higher molecular similarity leads to higher prediction reliability. However, activity cliffs are common in structure-property relationships and may indicate that this assumption fails under certain conditions. Can the authors please address these limitations in the discussion, and perhaps propose future work that may mitigate them?

Response and disposition:

Thank you for providing suggestions that improve the quality of our manuscript. Following the reviewer's comment, we have added descriptions in the Discussion and Conclusion sections about limitations of our method related to activity cliffs and proposed potential approaches to mitigate these issues. The specific additions are as follows.

- **Second Paragraph:** (lines 355–368; pages 14–15) “In addition, activity cliffs (ACs), a phenomenon where structurally similar molecules exhibit significantly different activities [1], require particular caution, especially defining ADs based on structure similarity-based metrics, such as MTS. In situations where ACs occur, the assumption that structural similarity correlates with higher prediction reliability may collapse and the defined AD may not work as intended. A tentative approach to address this issue is to include occurrence of ACs as part of the criteria for defining ADs. Significant efforts have been made to predict whether a given pair of molecules may exhibit ACs [2–7], and these methods could potentially be utilized to address the issue. A simple approach to handling ACs using these methods would be to classify molecule pairs predicted to cause ACs as outside ADs. Although it is currently challenging to completely avoid prediction errors caused by ACs using machine learning-based AC detection methods [8], addressing these issues remains an important area for future investigation.”
- **Fourth Paragraph:** (lines 385–391; page 15; A highlighted sentence is inserted.) “Incorporating the estimation of physical properties by quantum chemistry, all-atom, coarse-grained, or docking simulations into molecular design is also a promising strategy. **These methods are based on physicochemical principles and thus could mitigate risks associated with prediction reliability and activity cliffs which are inherent in data-driven property estimation.** Indeed, experimental validation of functional molecules designed by quantum chemistry calculations has been reported.”

References

1. Maggiora, G. M. On Outliers and Activity Cliffs – Why QSAR often Disappoints. *Journal of Chemical Information and Modeling* **46**, 1535–153 (2006).
2. Heikamp, K., Hu, X., Yan, A. & Bajorath, J. Prediction of activity cliffs using support vector machines. *Journal of Chemical Information and Modeling* **52**, 2354–2365 (2012).
3. de la Vega de León, A. & Bajorath, J. Prediction of compound potency changes in matched molecular pairs using support vector regression. *Journal of Chemical Information and Modeling* **54**, 2654–2663 (2014).
4. Husby, J., Bottegoni, G., Kufareva, I., Abagyan, R. & Cavalli, A. Structure-based predictions of activity cliffs. *Journal of Chemical Information and Modeling* **55**, 1062–1076 (2015).

- Horvath, D., Marcou, G., Varnek, A., Kayastha, S., de la Vega de León, A. & Bajorath, J. Prediction of activity cliffs using condensed graphs of reaction representations. *Journal of Chemical Information and Modeling* **56**, 1631–1640 (2016).
- Iqbal, J., Vogt, M. & Bajorath, J. Prediction of activity cliffs on the basis of images using convolutional neural networks. *Journal of Computer-Aided Molecular Design* **35**, 1157–1164 (2021).
- Park, J., Sung, G., Lee, S., Kang, S. & Park, C. ACGCN: graph convolutional networks for activity cliff prediction between matched molecular pairs. *Journal of Chemical Information and Modeling* **62**, 2341–2351 (2022).
- van Tilborg, D., Alenicheva, A. & Grisoni, F. Exposing the Limitations of Molecular Machine Learning with Activity Cliffs. *Journal of Chemical Information and Modeling* **62**, 5938–5951 (2022).

Comment 5:

The authors occasionally fail to distinguish between true properties and predicted properties. This distinction is important in this manuscript, as property predictions are being optimized in an effort to arrive at compounds with decent experimental properties. In line 112, for example, “each property” should be replaced with “each property prediction”.

Response and disposition:

Thank you for highlighting this point. We have revised the manuscript to explicitly distinguish between true and predicted properties throughout. The specific revisions are listed below:

- (line 112; page 3) “each property” -> “each property **prediction**”
- (bottom left of Figure 1d) “Set reliability level for each property” -> “Set reliability levels for each property **predictions**”
- (line 122; page 5) “setting the reliability level of each property” -> “setting the reliability level of each property **prediction**”
- (line 144; page 5) “the optimization of the properties may fail.” -> “the optimization of the **predicted** properties may fail.”
- (Fig. 2; caption; page 6) “DyRAMO explores an appropriate combination of the reliability level for each target property.” -> “DyRAMO explores an appropriate combination of the reliability level for each target property **prediction**.”
- (Fig. 2; caption; page 6) “a reliability level is selected for each property” -> “a reliability level is selected for each property **prediction**”
- (Fig. 2; caption; page 6) “the reliability levels given in step 1 for all properties” -> “the reliability levels given in step 1 for all **property predictions**”
- (lines 166–167; page 7) “To optimize multiple properties within the ADs of multiple prediction models”-> “To optimize multiple **predicted** properties within the ADs of multiple prediction models”
- (lines 175–176; page 7) “Weight w is allocated to each property for prioritization during optimization.” -> “Weight w is allocated to each **predicted** property for prioritization during optimization.”

10. (lines 179–180; page 7) “If the MTS between the designed molecule and training data exceeds the reliability level defined for all target properties,” -> “If the MTS between the designed molecule and training data exceeds the reliability level defined for all target **property predictions**,”
11. (lines 189–190; page 7) “we aim to achieve multi-objective optimization of the molecular properties within the ADs of multiple prediction models.” -> “we aim to achieve multi-objective optimization of the molecular **predicted** properties within the ADs of multiple prediction models.”
12. (lines 200–201; page 7) “The reliability level of each property was standardized by Scaler in the calculation of the DSS score.” -> “The reliability level of each property **prediction** was standardized by Scaler in the calculation of the DSS score.”
13. (lines 229–230; page 8) “with the other two properties also generally well optimized” -> “with the other two **predicted** properties also generally well optimized”
14. (lines 281–282; page 9) “adjusting the reliability level of each property” -> “adjusting the reliability level of each property **prediction**”
15. (lines 444–446; page 17) “The search spaces, consisting of combinations of reliability level value sets for each target property, are defined as follows” -> “The search spaces, consisting of combinations of reliability level value sets for each target property **prediction**, are defined as follows”

Remarks on code availability:

The Github repository appears to be well-organized and contains clear directions for running DyRAMO. DyRAMO's code was not tested as part of this review.

Response and disposition:

We are grateful for the reviewer's acknowledgment of the repository's clarity. While the code was not tested in this review, we have taken steps to ensure its reproducibility and ease of use.